

# Prevalence and richness of malaria and malaria-like parasites in wild birds from different biomes in South America

Daniela de Angeli Dutra[1,*], Nayara Belo[2,*] and Erika M. Braga[2]

[1] Department of Zoology, University of Otago, Dunedin, New Zealand
[2] Department of Parasitology, Universidade Federal de Minas Gerais, Belo Horizonte, MG, Brazil
* These authors contributed equally to this work.

## ABSTRACT

South America has different biomes with a high richness of wild bird species and Diptera vectors, representing an ideal place to study the influence of habitat on vector-borne parasites. In order to better understand how different types of habitats do or do not influence the prevalence of haemosporidians, we performed a new analysis of two published datasets comprising wild birds from the Brazilian Savanna (Cerrado) as well as wild birds from the Venezuelan Arid Zone. We investigated the prevalence and genetic diversity of haemosporidian parasites belonging to two genera: *Plasmodium* and *Haemoproteus*. We evaluated data from 676 wild birds from the Cerrado and observed an overall prevalence of 49%, whereas, in the Venezuelan Arid Zone, we analyzed data from 527 birds and found a similar overall prevalence of 43%. We recovered 44 lineages, finding *Plasmodium* parasites more prevalent in the Cerrado (15 Plasmodium and 12 Haemoproteus lineages) and *Haemoproteus* in the Venezuelan Arid Zone (seven *Plasmodium* and 10 *Haemoproteus* lineages).

No difference was observed on parasite richness between the two biomes. We observed seven out of 44 haemosporidian lineages that are shared between these two distinct South American biomes. This pattern of parasite composition and prevalence may be a consequence of multiple factors, such as host diversity and particular environmental conditions, especially precipitation that modulate the vector's dynamics.

The relationship of blood parasites with the community of hosts in large and distinct ecosystems can provide more information about what factors are responsible for the variation in the prevalence and diversity of these parasites in an environment.

# INTRODUCTION

Infectious and parasitic diseases can be considered a major threat to the conservation of native wild species and the emergence of diseases in wild environment has been attributed to various factors, including anthropization (*Patz et al., 2000*). It is known that the impacts caused to the environment in different regions of the world lead to an increase of wildlife exposure to vectors and parasites causing a significant loss of biodiversity (*Daszak, Cunningham & Hyatt, 2000*; *Jones et al., 2008*; *Paaijmans et al., 2010*; *Sehgal, 2015*). Environmental conditions of distinct biomes allow comparative studies on

Corresponding author
Erika M. Braga,
embraga@icb.ufmg.br

host-parasite interactions. More specifically, this comparative analysis can ensure a better understanding of the differences in the prevalence, diversity, and distribution of parasites in wild birds (*Sehgal, 2015*; *Clark, Drovetski & Voelker, 2020*; *Fecchio et al., 2021*). Certainly, avian malaria and malaria-like (haemosporidian) parasites have become an excellent model for the investigation of ecological dynamics of parasites in wildlife (*Marzal, 2012*). The genus *Plasmodium* and its phylogenetically related genus, *Haemoproteus*, are considered the most prevalent and diverse groups of parasites causing avian diseases. These parasites infect red blood cells and cells of the reticuloendothelial system of the vertebrate hosts and can cause severe acute infections on susceptible hosts (*Valkiūnas, 2005*).

The prevalence and distribution of haemosporidian parasites are intrinsically associated with the presence of susceptible vectors (*i.e.*, hematophagous Diptera), whose distribution is directly influenced by climatic conditions and habitat characteristics (*Sehgal, 2010*; *Loiseau et al., 2012*; *Krama et al., 2015*). Further, haemosporidian prevalence is also subject to biotic factors such as host behavior, age, sex, migration patterns and immune responses (*Gutiérrez-López et al., 2019*; *Ellis et al., 2020*; *de Angeli Dutra et al., 2021a*). For instance, the absence of blood parasites or even their low prevalence in resident birds at high altitudes or low temperatures regions and even in the marine environment can be explained by the lack or absence of competent vectors (*Hellgren, Bensch & Malmqvist, 2008*; *Krams et al., 2012*; *Martínez de La Puente et al., 2013*). Therefore, it is expected distinct biomes and environment to present particular and distinct parasite population dynamics.

Hence, since the interaction between biotic and abiotic factors drives haemosporidian transmission dynamics, abundance, and diversity of these microorganisms, we aimed to describe the differences on the prevalence and richness of *Plasmodium*/*Haemoproteus* parasites between two biomes, the Brazilian Cerrado and the Venezuelan Arid Zone in South America using previously published data (*Rodríguez-Ferraro & Blake, 2008*; *Belo et al., 2011*, *2012*). The Cerrado is the second largest biome in South America and is considered one of the world's biodiversity hotspots. The Cerrado has more than 800 bird species, of which nearly 20 are endemic and some species are listed as critically endangered on the IUCN Red List. Its climate is typical of a humid savanna region with mean temperatures around 25 °C and annual rainfall around 1,722 mm, which mostly occurs from October to April (rainy season) (*Viola et al., 2014*). On the other hand, the Venezuelan Arid Zone in the North of Venezuela is characterized by thorn scrubs dominated by species belonging to the Cactaceae, Mimoseae (Fabaceae), and Capparaceae taxa (*Sarmiento, 1976*). Moreover, the Venezuelan Arid Zone is characterized as an area of constant hydrologic shortage mainly because of drought and the influence of the trade winds of the Northern Hemisphere. The Venezuelan Arid Zone is an area of endemic specialist wild birds showing a low species richness (*Rodríguez-Ferraro & Blake, 2008*). The mean annual temperature is 28 °C, and annual rainfall ranges between 300 and 700 mm, with an extended and severe dry season punctuated by two brief rainy peaks in July–August, and December (*Sarmiento, 1976*). The characterization of the avian haemosporidian prevalence and diversity presented here is crucial for further studies addressing the host-parasite systems in neotropical environments.

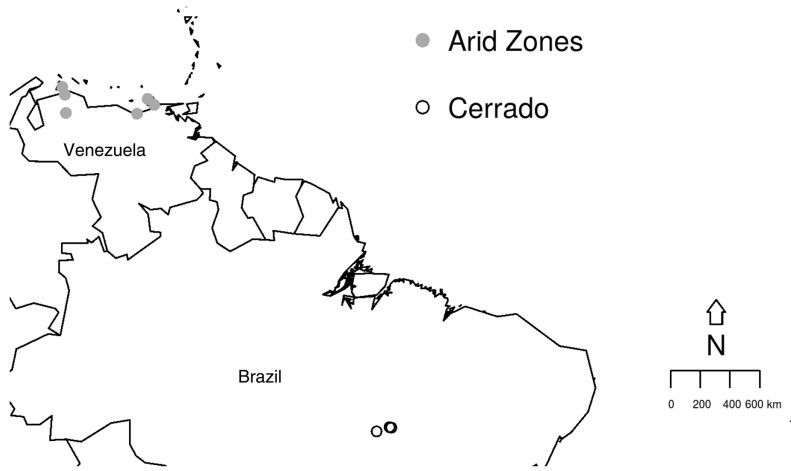

**Figure 1 Studied sites in the Brazilian Cerrado and the Venezuelan Arid Zone.** Map illustrating the collection sites within the two biomes studied: Cerrado (empty circles) and Venezuelan Arid Zone (grey circles).

## METHODS

### Study sites and blood sampling

The study aims to compare the prevalence and richness of haemosporidians from two different biomes depicting peculiar habitat characteristics using previously published data, as detailed below. The studied sites are (i) the Cerrado comprising three different areas in the Brazilian North region, Tocantins state (Fig. 1): an urban area in Palmas (10°10′S, 48°19′W) and, two protected areas, Lajeado (10°6′S, 48°14′W) and Cantão (10°26′S, 49°10′W) State Parks (*Belo et al., 2011*); (ii) six areas from the Venezuelan Arid Zone which had been previously studied (*Rodríguez-Ferraro & Blake, 2008*; *Belo et al., 2012*) (Fig. 1): Araya (10°34′N, 64°15′W), Macanao (10°58′N, 64°10′W) and Paraguaná (11°55′N, 70°02′W) Peninsulas, Falcon and Lara lowlands (11°23′N, 69°40′W) and, Clarines-Piritu (9°56′N, 65°9′W). In both previous published studies, molecular procedures for parasite identification were performed using amplification protocol described by *Ricklefs et al. (2005)*. Later, sequences were compared with data previously published in GenBank database. Data collection and molecular analyses were detailed in *Belo et al. (2011)* and *Belo et al. (2012)*. Field experiments were approved by Universidade Federal de Minas Gerais Ethics Committee for Experimentation (#205/2006), IBAMA/ ICMBio Brazil (#12322-3), Institutional Animal Care and Use Committee at UMSL/USA (Protocol 03-22) and, Inparques (N° 087) and the Dirección de Fauna at the Ministerio del Ambiente in Venezuela (PAA-035.2005).

### Statistics

In order to compare haemosporidian prevalence and richness between the studied biomes, we used Generalized Linear Mixed Models (GLMM) using the function "glmer" from "lme4" in R (*Bates et al., 2015*). For our first model, we evaluated the scaling of biome (reference level = Venezuelan Arid Zone) and season (reference level = dry season) on parasite infection status while controlling for locality and host species and family, which
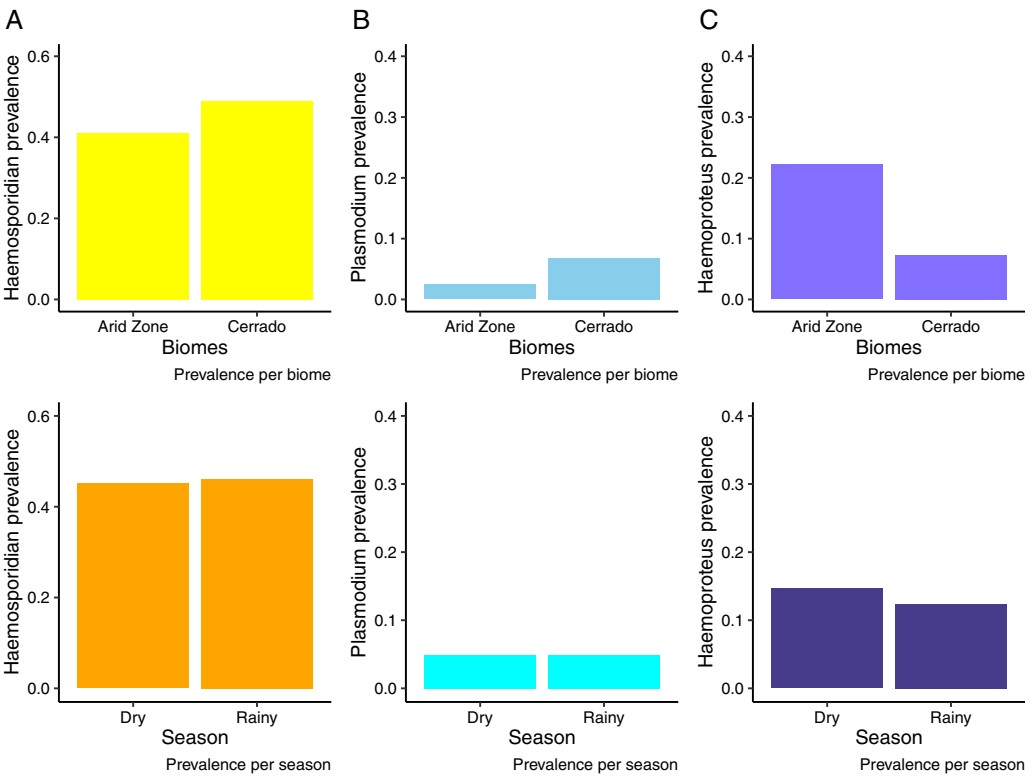

**Figure 2** **Haemosporidian prevalence.** Haemosporidian parasite prevalence among wild birds from two South American biomes (Brazilian Cerrado and Venezuelan Arid Zone) and in different seasons. (A) All haemosporidians; (B) *Plasmodium* lineages only; (C) *Haemoproteus* lineages only.

were added to the model as interacting random effects. Here, we employed a binomial distribution since our data analyses compared infected *vs*. non-infected hosts. We repeated the same analyses for *Plasmodium* and *Haemoproteus* parasites separately aiming to assess any distinct correlations associated with each parasite genus. Later, in order to compare parasite richness between the biomes, we grouped our dataset per locality. We analyzed the influence of biome on parasite richness running a Generalized Linear Model (GLM) with Poisson distribution and added total number of birds, prevalence, and bird richness as fixed factors. For this, we applied the basic stats R function "glm". Plots were created using "ggplot2" package in R. In order to obtained model estimates, we employed the basic stats function "summary" in R.

## RESULTS

We analyzed samples from 676 birds from the Cerrado comprising 122 species and 29 families evidencing an overall *Plasmodium/Haemoproteus* prevalence of 49% (range 42–56%) (331 birds infected). Whilst, in the Venezuelan Arid Zone we analyzed samples from 527 birds (20 species, 11 families), obtaining an overall prevalence of 43% (range 36–53%) (Fig. 2). Tables S1 and S2 list bird species and families sampled and infected by haemosporidian parasites. We observed that birds from the Cerrado presented higher prevalence of haemosporidians when compared to the Venezuelan birds analyzed (Fig. 2A,

Table 1 Parameter estimates, standard errors, z and *p*-values for the GLMM testing the relationship between haemosporidian prevalence and biome and seasonality.

|  | Estimate | Standard Error | z-value | *p*-value |
|---|---|---|---|---|
| *Intercept* | −0.486 | 0.320 | −1.519 | 0.128 |
| *Biome* | 0.84 | 0.306 | 2.761 | 0.005 |
| *Season* | −0.15 | 0.15 | −0.996 | 0.319 |

Table 2 Parameter estimates, standard errors, z and *p*-values for the GLMM testing the relationship between *Plasmodium* prevalence and biome and seasonality.

|  | Estimate | Standard Error | z-value | *p*-value |
|---|---|---|---|---|
| *Intercept* | −4.087 | 0.480 | −8.454 | <0.001 |
| *Biome* | 1.211 | 0.494 | 2.450 | 0.014 |
| *Season* | −0.374 | 0.309 | −1.210 | 0.226 |

Table 3 Parameter estimates, standard errors, z and *p*-values for the GLMM testing the relationship between *Haemoproteus* prevalence and biome and seasonality.

|  | Estimate | Standard Error | z-value | *p*-value |
|---|---|---|---|---|
| *Intercept* | −1.806 | 0.386 | −4.677 | <0.001 |
| *Biome* | −1327 | 0.435 | −3.047 | 0.002 |
| *Season* | 0.236 | 0.226 | 1.043 | 0.297 |

Table 4 Parameter estimates, standard errors, z and *p*-values for the GLM testing the relationship between haemosporidian richness and biome.

|  | Estimate | Standard Error | z-value | *p*-value |
|---|---|---|---|---|
| *Intercept* | 2.212 | 0.756 | 2.808 | 0.005 |
| *Biome* | −1.197 | 2.463 | −0.486 | 0.626 |
| *Number of birds per locality* | −0.005 | 0.006 | −0.800 | 0.423 |
| *Bird richness* | 0.054 | 0.070 | 0.779 | 0.436 |
| *Prevalence* | −0.686 | 2.463 | −0.486 | 0.733 |

Table 1). Nevertheless, when analyzing parasite genera separately, we found out that the prevalence varies between parasite genera. *Plasmodium* parasites were more prevalent in the Cerrado, with 6% (range 5–10%) of prevalence against 2% (range 0–5%) in the Venezuelan Arid Zone. On the other hand, *Haemoproteus* parasites reached higher prevalence rates in the Venezuelan Arid Zone, with 21% (range 12–33%) against 7% (range 5–9%) of prevalence in the Cerrado (Figs. 2B and 2C, Tables 2 and 3). No difference in haemosporidian prevalence was detected between seasons (Tables 1–3). Likewise, parasite richness is similar between both biomes and does not seem to be driven by either parasite prevalence or bird abundance and richness (Table 4). It is important to notice that this study evaluates a limited number of localities.

Sequencing of the cytochrome *b (cyt b)* gene revealed 22 *Plasmodium* lineages and 22 *Haemoproteus* lineages. Of 44 lineages, 27 lineages were observed in the Cerrado (15 *Plasmodium* lineages and 12 *Haemoproteus* lineages) while 17 parasites lineages were found in the Venezuelan Arid Zone (7 *Plasmodium* lineages and 10 *Haemoproteus* lineages). We observed seven identical lineages in the Cerrado and Arid biomes: *Plasmodium* lieneages Ven7 identical to Toc4, Ven1 identical to Toc15, Ven15 identical to Toc32 and, *Haemoproteus* lineages Ven13 identical to Toc20, Ven16 identical to Toc1, Ven12 identical to Toc3, and Ven2 identical to Toc2 (Ven = Venezuela and Toc = Tocantins). The *Haemoproteus* Toc2 and *Haemoproteus* Ven2 lineages are molecularly similar and harbor a higher number of host families, comprising 13 different host families. This parasite lineage is widespread and has been previously reported in different regions of the globe (*e.g.*, Colombia, Peru, and Russia). On the other hand, we observed a total of eight different parasite lineages infecting multiple host families in the Cerrado. Thirty-one of the lineages recovered in our study had been previously described and are considered widespread. The greatest number of widespread lineages (14) was found in the Cerrado. Seven distinct lineages from each biome were local (*i.e.*, not found in any other areas).

## DISCUSSION

In order to better understand how different types of habitats can or not influence the prevalence of haemosporidians, we analyzed wild birds from the Brazilian Cerrado and from the Venezuelan Arid Zone. Here, 44 *Plasmodium/Haemoproteus* lineages were observed, 27 of which were detected in the Cerrado and 17 in the Venezuelan Arid Zone. Of these, we observed that seven lineages (three lineages of *Plasmodium* and four lineages of *Haemoproteus*) were recovered from the Brazilian Cerrado as well as from the Venezuelan Arid Zone birds. Moreover, those lineages were detected in several host species from different bird families, which may indicate that these lineages are generalists and capable of infecting different birds in distinct environmental conditions.

*Plasmodium* and *Haemoproteus* parasites may be dispersed through migration (*Hellgren et al., 2007*; *de Angeli Dutra et al., 2021b*). For instance, *Waldenstrom et al. (2002)* detected haemosporidian lineages in many African samples that had also been detected in populations of migratory birds that breed in Europe. In South America, migratory species of the genus *Tyrannus* circulate between Brazil and Venezuela. The Tyrannidae represent about 33.5% of the birds that perform this type of displacement (*Remsen, 1997*). The subspecies *Myiachus swainsoni swainsoni* (species infected by *Plasmodium* in the arid region of Venezuela in this study), breeds in the southernmost regions of South America and winters in the north of this continent. During the austral winter, June–August, another Tyrannidae (*Elaenia parvirostris*) moves north across the Amazon, wintering in Colombia, Venezuela, and the Guianas (*Remsen, 1997*). Nonetheless, even though migrants can disperse haemosporidians, most of their diversity is harbored by the resident bird community (*de Angeli Dutra et al., 2021b*). Hence, migration may possess a limited role on the sharing of parasites between the Cerrado and the Venezuelan Arid Zone.

On the other hand, when a host species is abundant there is a greater opportunity for parasites that are specialists to prevail than generalists at local scale (*Poulin & Mouillot, 2004*). However, when there is a lower abundance of different hosts, the transmission of parasite species between hosts of the same species is lower and generalist parasites prevail (*Keesing, Holt & Ostfeld, 2006*). Indeed, generalist *Plasmodium* and *Haemoproteus* parasites are considered better colonizers than specialists (*de Angeli Dutra, Moreira Félix & Poulin, 2021*). Differences in the quantity and number of host species, as well as vector transmission and ecology, seem to be more important in structuring *Plasmodium/Haemoproteus* communities than the distance between different habitats (*Ishtiaq et al., 2009*; *Svensson-Coelho & Ricklefs, 2011*). Therefore, one can expect lineages that can reproduce in heterogenous habitats and hosts could be highly dispersed throughout the globe. This pattern has already been observed in other studies, in which no effects of geographic distance on the composition of *Plasmodium/Haemoproteus* parasites were observed (*Scordato & Kardish, 2014*).

Dissimilarity between parasite communities can be driven by a turnover on host composition (*de La Torre et al., 2021*), which may occur because of host specialization and, consequently, cophylogenetic congruence. Avian haemosporidians seem to have coevolved and diversified following their hosts phylogeny (*Pacheco et al., 2018*; *de Angeli Dutra et al., 2022*). Therefore, distinct bird communities are likely to harbors unlike parasites. Here, we observed seven out of 44 haemosporidian lineages that are shared between two distinct South American biomes and a distinct prevalence pattern with *Plasmodium* parasites being more prevalent in the Brazilian Cerrado and *Haemoproteus* in the Venezuelan Arid Zone. This difference on parasite taxonomic composition and prevalence, however, can be a consequence of multiple other factors, such as the dissimilar environmental conditions, especially precipitation. Since *Plasmodium* and *Haemoproteus* parasites utilize different vectors for their development, distinct environmental conditions can favor the development of one parasite taxa over another. For instance, since mosquitoes require higher precipitation rates to reproduce (*Forattini, 1995*), our results may reflect this shift in vector abundance between the two regions evaluated. At same time, we found similar prevalence rates between seasons in all our models, indicating birds are either infected through the year evenly or maintain detectable chronical infections during the entire year.

Correlating the dynamics of blood parasites with the community of hosts and vectors in large and distinct ecosystems can provide more information about which factors are responsible for the variation in the prevalence and diversity of these parasites in an environment. There is a need to carry out more studies in Neotropical regions and in different habitats to better understand the relationships between the environment, the abundance and diversity of vectors and parasites and their respective effects on the host.

## ACKNOWLEDGEMENTS

We are grateful to Dr Robert Ricklefs and Dr Adriana Rodríguez-Ferraro for providing data from birds and parasites from Venezuela.

## Funding

This work was supported by CNPq (Conselho Nacional de Desenvolvimento Científico e Tecnológico–Grant 501519/2013-0; Grant 304334/2019-7), CAPES (Coordenação de Pessoal de Nível Superior-Grant 23038.005277/2011-68) and, Fundação de Amparo à Pesquisa do Estado de Minas Gerais (FAPEMIG–Grant BPD-00122-14). Daniela de Angeli Dutra was supported by the University of Otago Doctoral Scholarship. The funders had no role in study design, data collection and analysis, decision to publish, or preparation of the manuscript.

## Grant Disclosures

The following grant information was disclosed by the authors:
CNPq (Conselho Nacional de Desenvolvimento Científico e Tecnológico): 501519/2013-0 and 304334/2019-7.
CAPES (Coordenação de Pessoal de Nível Superior): 23038.005277/2011-68.
Fundação de Amparo à Pesquisa do Estado de Minas Gerais (FAPEMIG): BPD-00122-14.
University of Otago Doctoral Scholarship.

## Competing Interests

Erika M. Braga is an Academic Editor for PeerJ.

## Author Contributions

- Daniela de Angeli Dutra conceived and designed the experiments, performed the experiments, analyzed the data, prepared figures and/or tables, authored or reviewed drafts of the paper, and approved the final draft.
- Nayara Belo conceived and designed the experiments, performed the experiments, analyzed the data, prepared figures and/or tables, authored or reviewed drafts of the paper, and approved the final draft.
- Erika M. Braga conceived and designed the experiments, analyzed the data, authored or reviewed drafts of the paper, and approved the final draft.

## Animal Ethics

The following information was supplied relating to ethical approvals (*i.e.*, approving body and any reference numbers):

Universidade Federal de Minas Gerais Ethics Committee for Experimentation: Permit Number: 205/2006.

Institutional Animal Care and Use Committee at UMSL/USA (Protocol 03-22).

Inparques (N° 087) and the Dirección de Fauna at the Ministerio del Ambiente in Venezuela (N° PAA-035.2005).

## Field Study Permissions

The following information was supplied relating to field study approvals (*i.e.*, approving body and any reference numbers):

Field experiments were approved by Universidade Federal de Minas Gerais Ethics Committee for Experimentation, IBAMA/ICMbio, Brazil Committee at UMSL/USA and, Inparques and the Dirección de Fauna at the Ministerio del Ambiente in Venezuela.

## DNA Deposition

The following information was supplied regarding the deposition of DNA sequences:

The sequences are available at GenBank: HQ287536–HQ287556 and JN819517–JN819533.

## Data Availability

The raw data are available in the Supplemental Files.

## Supplemental Information

Supplemental information for this article can be found online at http://dx.doi.org/10.7717/peerj.13485#supplemental-information.

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
