# Peer review of "Prevalence and richness of malaria and malaria-like parasites in wild birds from different biomes in South America"

_PeerJ, doi:10.7717/peerj.13485_

## Round 0.1 · original submission · Major Revisions

Dear colleagues,

I have received comments from two expert reviewers for your manuscript. While both (and myself) agree that this is an interesting and valuable study, they indicate a number of modifications are required before publication.

I advise the authors to address all comments made by the reviewers, but I would like to draw special attention to the following aspects:

1. Both reviewers and myself interpret that this study is a data analysis from previously obtained empirical data by some of the authors (Belo et al. 2011 and 2012). If this is the case (as I assume it is, because I have checked some of the GenBank accession numbers and they indeed refer to the Belo et al. 2011 paper), then it needs to be clarified in methods, instead of descring all methodological procedures for DNA extraction, sequencing, etc. If this is not the case, if extra molecular data has been obtained, then please clarify;

2. There is some uncertainty about the source of genetic data, especially in the section "phylogenetic methods", as reviewer #2 pointed out. I suspect this is related to issue number 1 that I have raised here. It seems to me that the cytb data is in fact comparable to MalAvi, but it is not clear to what extent and exactly how the authors performed this comparison. Additionally, reviewer #2 is correct in that the section does not describe any phylogenetic methodology at all -- which leaves a question on how these sequences were treated in order to determine their identity. Were there phylogenetic trees or was identity defined by just blast? If just blast, what were the thresholds and other parameters? It may be the case that this methodology is described in the previous papers, which once again emphasizes the importance of clarifying exactly what the origin of the data is -- it is not necessary to repeat the methodological description here, but it would be important to clarify these aspects.

I look forward to receiving a revised version of your manuscript.

·

Basic reporting

This manuscript is overall clearly structured and well written, except for a few ambiguous sentences (see specific comments). The background is clear and concise. However, the authors should make clear in the introduction that this study is merely a comparison of data et results published in two other research articles (Belo et al 2011, Belo et al 2012). I think it is a good thing to add value to published data, but it should be clearly stated, which is not the case here. These two papers are not cited in the introduction, and one of them (Belo et al. 2011, for the Brazilian data) is even missing in the reference list. It is only briefly mentioned in the Study site section of the material and methods that the “areas had been previously studied”, which is rather ambiguous. The fact that the authors describe entirely the data collection, the wet lab and phylogenetic analyses is misleading because it looks like it is new work. I think the authors should refer to the published papers (from which the data come from) for most of the methods, and only describe in detail the new statistical analyses they performed (and other potential new additions).
The figures are relevant and correctly labelled.
Suggestions for Figure 1: perhaps writing countries’ names (at least Brazil) on the map would make it more immediately readable (and the scale and the North arrow would not be necessary). And, if not too fastidious, using a pin-shaped symbol would improve clarity for the Brazilian sampling sites.
Figure 2: what do the error bars represent? If space allows, here is a suggestion: for each plot, it may be nice to show a bar for each sampling site (grouped by biomes) and a line over of the bars to represent the mean for the biome (instead of only the means per biome). It would also be interesting – again, if space allows - to see parasite community composition plots for each sampling site.
For potential meta-analysis, it would be useful to deposit raw data in the form of a .csv or .txt table (with species, sampling date, sampling coordinates, infection status and parasite sequence when applicable and any other interesting variable that was collected for each captured individual).

Experimental design

The GLMMs sound appropriate to compare prevalence and parasite richness of these two biomes.
L.150: Authors should cite the publications linked to the R packages they used (Bates, D., Mächler, M., Bolker, B., & Walker, S. (2015). Fitting linear mixed-effects models using lme4. Journal of Statistical Software, 67, 1–48. https://doi.org/10.18637/jss.v067.i01).
L.152: if the response variable was 1/0 (infected/uninfected) each bird, I would rather use “infection status” than “prevalence”, as prevalence refers to a proportion of infected individuals in a population/group.
L.156: I guess it is a Poisson, and not poison, distribution
L.157: if I understand correctly, you built a model with the species richness per locality as a response variable, and locality as a random effect (and biome, prevalence, number of birds and bird richness as fixed variables). I do not think that a model that has a different random effect value for each response variable value is supposed to converge. Maybe I am missing something, but I do not think you should use locality as random effect.
In addition, you could test whether species richness varies with seasons. These data would also allow for analyses on species compositions.
I will not comment on the rest of the method section as I think it should be removed and reference to the original papers should be made, but I have one question: were the birds ringed to avoid sampling some individuals several times?
If the authors do not remove the rest of the method section, they should at least describe how were treated mixed infections and report the number of positive samples for which they were unable to identify lineages.

Validity of the findings

The result section should only report results that have not been published yet, that is the results of the GLMMs (if I understood correctly). Otherwise, the conclusions are well-stated, linked to the original research question.

Additional comments

As PeerJ is a generalist journal, it might be worth mentioning that Plasmodium causes avian malaria.
Other specific comments:
L.55: not only Culicidae, but also Ceratopogonidae and Hippoboscidae
L.63: not sure about this, but I think “infection dynamics” refers to what happens within an infected host. To avoid ambiguity, I would use “parasite population dynamics”.
L.65: “transmission dynamics, abundance and diversity”
L.87: punctuation
L.164: haemosporidian parasites
L.180: on what criteria are these lineages similar? Were the sequences similar?
L.186: were these seven lineages never found in any other area, or only in your sampling sites?
L.198-200: This sentence repeats itself.
L.203-204: “The Tyrannidae represent about 33.5% of the birds that perform this type of displacement (Remsen, 1997).”

Reviewer 2 ·

Basic reporting

The article is well written.

Line 58: I recommend replacing “strategies” with “response”.
Line 65: include a comma after dynamics and delete “and”.
Lines 88-89: include a comma after Palmas. Please, include geographic coordinates information about study sites, both in Brazil and Venezuela.
Line 117: I believe that the authors would like to say “1.5%”, please, check along with the text.
Lines 121-129: Please, check the space between numbers and scale. For example “20sec” or “20 sec”, “68°C” or “68 °C”. Check along the text and standardize. Please, explain the abbreviation “cytb”, and check how it is written along with the text.
Line 144: Why do the authors mean by “pub- listed”?
Line 161: I recommend including “Plasmodium/Haemoproteus” before overall.
Line 165: Say “We observed that the birds from …”
Line 178: check the spell of lineages
Line 190: delete “the Brazilian”
Line 195-196: include “that” after indicate. Include “and” after generalists.

Figure 1: I recommend including that the Cerrado is in Brazil and that the Aride Zone is in Venezuela.
Figure 2: Please, include which South American biomes are being studied.

Tables 1-4: Use the same spelling of “z-value” and “p-value” in the table title and in the table. I also believe that tables 1-3 can be merged into one.

Supplementary tables: Maybe these tables can be merged into one. Additionally, it would be interesting to have GenBank accession numbers on it, the lineage that was found, and if they were infected by Plasmodium or Haemoproteus.

Experimental design

The manuscript “Prevalence and richness of haemosporidians in wild birds from different biomes in South America” is well written and presents interesting data about the diversity of parasite lineages in the studied places. However, I miss several information and discussions that I feel would bring more interesting discussions to this manuscript.

First, were these samples collected and processed exclusively for this study? In lines 88-92 I had the impression that the samples mentioned in the present study were collected, processed, and published before. If that’s the case, then I don’t see the point of showing sections such as DNA extraction, screening, cytochrome b amplification and sequencing, and phylogenetic analysis.

Second, the authors are dealing with haemosporidian parasites, widely studied by several research groups all over the world. In order to standardize the name of lineages (haplotypes) and to keep all data about the host and distribution of these lineages, and haemosporidian database was created MalAvi, which I’m pretty sure the authors know about it. However, the authors decided to amplify a DNA fragment that does not completely overlaps the fragment used in MalAvi. Using this fragment would allow us to determine which parasite lineages (and even species) were found in both studied biomes? Said that, why did the authors decide to use a different fragment for this study?

Third, this manuscript does not show any phylogenetic results or description of parasite lineage, the authors only compare sequences to know which of them were similar. Due to that, I would suggest editing the title of the subsection “Phylogenetic analysis” or even deleting it.

Validity of the findings

no comment

Additional comments

Lines 29-31: I suggest including the importance of bird species diversity as one of the important factors that interfere in parasite diversity.

Line 55: Culicidae is responsible for the transmission of Plasmodium only. Haemoproteus are transmitted by Ceratopogonia and Hippoboscidae. Leucocytozoon is transmitted by Ceratopogonidae and Simuliidae. Please, include this information.

Line 67: Sometimes the authors refer to “Venezuelan Arid Zone”, others as “arid zone”, “Arid Zone”, and “arid region of Venezuela”. Please, standardize the way it is written.

Line 70: IUCN does not use the term “highly threatened” in its classification. Maybe “Critically Endangered” would be more appropriate. Please, check the species that have such classifications and replace it with the correct term.

Lines 72-80: Authors mentioned the diversity of bird species, endemic bird species, and that some of them are threatened with extinction for the Brazilian Cerrado. I recommend including this information about Venezuelan Arid Zone.

Lines 98-100: Is there any official approval for the study in Venezuela. If yes, please include the numbers.

Lines 103-107: Authors are always talking about the Cerrado, and after that about the Arid Zone. I recommend continuing with this sequence along with the text. Please, review this paragraph. The same applies to the figure 1 legend.

Line 115: please, check the extra space at the end of the line.

Lines 178-179: I believe that “Ven” and “Toc” refer to the lineages found in Venezuela and Tocantins, respectively. Please, include this information in the manuscript, because when you say “Ven7-Toc14” it seems that this is only one lineage.

---

## Round 0.2 · Minor Revisions

Dear colleagues,

Thank you for the revised manuscript. As you will see, the reviewer and myself agree that the large majority of comments and critiques were addressed in satisfactory form.

The reviewer points out a few grammar/spelling mistakes that should be fixed, and also makes a couple of observations regarding the plots which I think you should take into consideration.

·

Basic reporting

Most suggestions have been included in the manuscript, which in my opinion almost passes the criteria. It is in general clearly written but there are some typos and inconsistencies left (for example regarding capital letters at the beginning of words), that could be corrected after a thorough reading. I am not a native English speaker, so I probably missed some, but here are a few suggestions:


L. 76: “using previously published data”

L. 84: “in northern Venezuela” or “in the North of Venezuela”

L. 99-100: “using previously published data, as detailed below”

L. 103: “State Parks (Belo et al., 2011);”

L. 108-110: I suggest removing this sentence, as sequence comparison was also described in Belo et al 2011 and Belo et al 2012, in a clearer way (“Sequences were compared for identification to their closest matches in GenBank using the NCBI nucleotide Blast search, and to unpublished sequences using a local blast search in the laboratory of R. E. Ricklefs.”).

L. 110-112: Rephrasing suggestion: “Data collection and molecular analyses were detailed in Belo et al. 2011 and Belo et al., 2012.”

L. 123: maybe “assess” instead of “access”

L. 126: “generalized linear model” (either remove capital letters here or add them in L. 116 “Generalized Linear Mixed Models”)

L. 137: “haemosporidian”

L. 152: “observed”

L. 153: “lineages”

L. 216: especially

Finally, I disagree with responses regarding the plots. As seasons and locality are used in GLMMs as fixed and random variables respectively, it seems relevant to illustrate which part of the intra-biome variability can be attributed to these variables. Adding the means per biome to such plot would do the same job than only showing simple barplots. However, I will not insist if authors disagree.

I do not understand why error bars were removed from the plots. If the authors do not want to illustrate prevalence per locality, it would at least be nice to have error bars showing standard deviation.

Experimental design

L. 129: I’m sorry I missed it during my first review, but please explain how you obtained p-values of the variables in the GLM(M)s.

Validity of the findings

OK

Additional comments

no comment

---

## Round 0.3 · accepted · Accept

Thank you for addressing all comments and critiques made by the reviewers. This manuscript is now ready for publication.